# The Chemistry and Biology of Cyclophostin, the Cyclipostins and Related Compounds

**DOI:** 10.3390/molecules24142579

**Published:** 2019-07-16

**Authors:** Christopher D. Spilling

**Affiliations:** Department of Chemistry & Biochemistry, University of Missouri-St. Louis, One University Boulevard, St. Louis, MO 63121, USA; cspill@umsl.edu

**Keywords:** cyclophostin, cyclipostins, enolphosphate, enolphosphonate, bicyclic, monocyclic, lipase, inhibitor

## Abstract

Cyclophostin, the cyclipostins and the salinipostins are structurally related cyclic enolphosphate natural products. This mini review describes their isolation, synthesis and biological activities. In addition, the synthesis and biological activities of monocyclic enolphosphate and mono and bicyclic enolphosphonate analogs are presented.

## 1. Introduction

Medicinal chemists have often turned to nature as a source of bioactive compounds, particularly for anticancer or antimicrobial agents [1,2,3]. Indeed, in a recent review it was reported that of the drugs approved between 1981 and 2010, 34% were based on natural products, either directly or as derivatives [3]. The phosphate moiety is ubiquitous throughout nature and therefore phosphate containing natural products should provide additional opportunities for drug discovery. It is therefore not surprising that over the past 30 over years some very interesting, structurally related, biologically active bicyclic enolphosphates have been isolated from various *Streptomyces* strains. Their isolation, synthesis and biological activities, and the synthesis and biological activities of monocyclic enolphosphate and mono and bicyclic enolphosphonate analogs are discussed below.

## 2. Discussion

### 2.1. Isolation and Structure of Cyclophostin

In 1987, Neumann and Peter reported the isolation of two organophosphates during a search for natural insecticides [4]. The compounds CGA 134,736 (**1a**) and CGA 134,735 (**1b**) (Figure 1) were isolated from a soil organism *Streptomyces antibioticus* DMS 1951. The structures were assigned by X-ray crystallography and synthesis, although no spectroscopic or structural details were provided and no follow-up publications have appeared. Both compounds were reported to be good inhibitors of acetyl cholinesterase (AChE) with IC_50_ of between 0.7 and 5.7 × 10^−7^ M.

Six years later, cyclophostin (**2a**) (Figure 1) was isolated from *Streptomyces lavendulae* (strain NK 901093) [5]. The structure was assigned using spectroscopic data and the absolute stereochemistry by X-ray crystallography with complete details of both reported in the manuscript. Cyclophostin is characterized by a unique cyclic phosphate triester fused to a lactone ring. The bicyclic core also contains an unusual vinylogous phosphate carbonic anhydride and has chiral centers at both phosphorus and the C-8a carbon atom (3*R*,8a*R*) (see Section 2.4 below) with the methoxy and ring junction hydrogen in a *cis* relationship. Cyclophostin is a potent inhibitor of AChE with IC_50_ of 7.6 × 10^−10^ M for the enzyme from housefly and 1.3 × 10^−9^ M for the enzyme from brown plant hopper.

### 2.2. Isolation of the Cyclipostins

In 2002, a family of structurally related natural products named the cyclipostins (**3a**–**j**) was reported (Figure 2). The cyclipostins were isolated from fermentation broths of *Streptomyces sp*. DSM 13381 [6,7,8]. Members of the cyclipostins family contain the same bicyclic core seen in cyclophostin, but vary in the nature of the lipophilic chain attached to the phosphate ester. The structures were assigned from spectroscopic data. Similar to cyclophostin, the cyclipostins possess the 3*R*,8a*R* relative stereochemistry, but no information on absolute configuration was given.

Many members of the cyclipostin family possess strong inhibitory activity against hormone-sensitive lipase (HSL) with IC_50_ values in the nanomolar range. Furthermore, the lipophilic cyclipostins were also shown to block lipolysis in intact rat adipocytes by direct inhibition of hormone-sensitive lipase (HSL) [7]. A comparison of cell-free and whole-cell activity showed that the cyclipostins are efficiently transported into the cell. Further studies also showed that cyclipostins possess anti-mycobacterial activity [9]. Although the mechanism of the anti-mycobacterial was not directly discussed, the patent referred to enzymes related to hormone sensitive lipase produced by mycobacterium tuberculosis.

### 2.3. Isolation of the Salinipostins

In 2015, the Salinipostins (**4**) (Figure 3) were isolated from a marine sediment organism [10]. They are structurally similar to the cyclipostins, but with the alkoxy (ester) and ring junction hydrogen in a *trans* relationship or (3*R*,8a*S*) (see Section 2.4). The structures were assigned based on spectroscopic data and comparison with compounds in the literature. Salinipostin A is a potent growth inhibitor of *Plasmodium Falciparum* (EC_50_ = 50 nM), the causative agent of malaria.

### 2.4. Stereochemical Nomenclature

In most cases, the structural determination for cyclophostin, the cyclipostins, and the salinipostins were made on the basis of spectroscopic data. However, determination of the structure of cyclophostin was also confirmed by X-ray crystallography. As pointed out by Spilling et al. in the first paper describing the synthesis of (±) cyclophostin and (±) cyclipostin P [11], the configuration named for cyclophostin in the isolation paper (3*R*,8a*S*) is incorrect. A common error is to treat P^+^–O^−^ as P=O and hence assign the group priority incorrectly [12]. Natural cyclophostin is actually 3*R*,8a*R*. {CA index name 1*H*,6*H*-Furo[3,4-e][1,2,3]dioxphosphepin-6-one, 8,8a-dihydro-3-methoxy-5-methyl-, 3-oxide (3*R*,8a*R*)}. The authors of the salinipostin isolation paper made the same error in naming the stereochemistry as *S_P_*,*S_C_* (3*S*,8a*S*), whereas it is in fact *R_P_*,*S_C_* (3*R*,8a*S*). In both cases, the configurations are correctly named in chemical abstracts.

### 2.5. Biosynthesis

Cyclophostin, the cyclipostins, and the salinipostins are structurally related bicyclic enolphosphates. Indeed, Salinipostin F has the same enolphosphate alkyl substituent (nPr) and phosphate ester of only one methylene unit less (C_15_) than cyclipostin T (C_16_), albeit as the opposite diastereoisomer. In addition, there appears to be a structural relationship of cyclophostin, the cyclipostins and the salinipostins to other natural products isolated from *Streptomyces* species, e.g., the virginiae butanolides and A-factor (Figure 4), suggesting that there may be a common biosynthetic origin [13,14,15,16,17,18].

### 2.6. Synthesis of (±) Cyclophostin and (±) Cyclipostin P

The synthesis of racemic cyclophostin (**2a**) and cyclipostin P (**3d**) was reported in 2011 [11]. The hydroxyl moiety of lactone (**5**), which is available as a racemic mixture or as either enantiomer, was protected as a *p*-methoxybenzyl (PMB) ether (**6**) using the copper (II) triflate-catalyzed reaction of *p*-methoxybenzyloxy trichloroacetimidate (Scheme 1).

The acetyl group was installed by deprotonation of the lactone with NaHMDS and acylation with acetic anhydride. Selective phosphorylation of the resulting 2-acetyl butyrolactone (**7**) to form the *E*-enol phosphate (**8**) was achieved by reaction with dimethyl chlorophosphate using an organic base. The PMB-ether was removed using DDQ in wet CH_2_Cl_2_ and the enolphosphate (**9**) was selectively mono-demethylated using one equivalent of sodium iodide in acetone at 45 °C. The sodium salt was protonated with Amberlite IR 120^®^ resin to generate the corresponding phosphoric acid (**10**). The phosphoric acid (**10**) was successfully cyclized with DCC and DMAP in CH_2_Cl_2_ to give the cyclic enolphosphates (**2a**) and (**2b**) as a 1:1 mixture. The diastereoisomers were separated using silica gel chromatography to give natural (±) cyclophostin (**2a**) and its diastereoisomer (**2b**) in 55% combined yield.

Cyclophostin (**2a**) and its diastereoisomer (**2b**) were converted to the cyclipostins by a novel one pot ester exchange process (Scheme 2). For example, the unnatural diastereoisomer **2b** was treated with hexadecyl bromide (10 equivalents) and catalytic tetrabutylammonium iodide (TBAI) in refluxing dioxane to give cyclipostin P (**3d**) and its diastereoisomer (**13**) in a 1:1 ratio with 95% conversion. The diastereoisomers were separated by column chromatography to give a 77% combined yield. The reaction of either (**2a**) or (**2b**) with hexadecyl bromide and catalytic TBAI in refluxing dioxane resulted in a similar 1:1 mixture of (**3d**) and (**13**).

### 2.7. Synthesis of the Salinipostins

Tao and coworkers reported the synthesis of six racemic salinipostins (**4**) and their diastereoisomers (**14**) (Scheme 1 and Scheme 2) [19]. They adopted the chemistry developed by Spilling et al. [11] for the synthesis of cyclophostin and the cyclipostins, substituting various carboxylic anhydrides for acetic anhydride to introduce the enol phosphate alkyl substituents (R^1^) required for the salinipostins. These researchers used EDC for the cyclization of the phosphoric acids.

### 2.8. Synthesis of Mono and Bicyclic Phosphonate Analogs

Spilling et al. reported the synthesis of phosphonate analogs of cyclophostin and the cyclipostins [20] and later adopted the chemistry to the preparation of several monocyclic phosphonate analogs [21,22,23,24]. The key C–C bond forming reaction involved a palladium-catalyzed substitution reaction of phosphono allylic carbonates (Scheme 3). The palladium-catalyzed reaction of methyl acetoacetate with the phosphono allylic carbonates (**15**) gave the vinyl phosphonates (**16**) in good yield. Selective hydrogenation of the vinyl phosphonate (**16**) using hydrogen over 10% Pd/C poisoned with pyridine gave the saturated phosphonates (**17**), which after selective demethylation, protonation of the resulting salt, and cyclization gave the monocyclic enolphosphonates (**18**). For the phosphonates, a combination of EDC and HOBt were the preferred reagents for cyclization. The *cis* and *trans* stereochemistry (OMe to C5-H) was assigned initially by X-ray crystallography on (**18h**) and then a comparison of the ^31^P NMR signals [22].

The phosphono allylic carbonates (**15**) can be prepared enantiomerically enriched and the palladium-catalyzed reaction of methyl acetoacetate proceeds with high transfer of the stereochemical information from the α position in the carbonate (**15**) to the γ position of the vinyl phosphonate (**16**). This allowed the preparation of both enantiomers of both diastereoisomers of (**18e**) with enantiomeric excess (e.e.) of >85% [23].

The bicyclic phosphonate analog of cyclophostin was available by the debenzylation of phosphonate (**18b**). Debenzylation of (**18b**) with hydrogen over palladium on carbon (Scheme 4) resulted in rapid lactonization to give the phosphonate isostere of cyclophostin (**19a**) and its diastereoisomer (**19b**) [20].

Conversion of the cyclophostin phosphonate analog (**19a**) to the diastereoisomeric cyclipostins analogs (**20a** and **20b**) was achieved via in situ selective cleavage of the methyl phosphonate ester with tetrabutyl ammonium iodide (TBAI) and re-alkylation with a long chain alkyl bromide (Scheme 5). This reaction sequence was also successful with the monocyclic phosphonate analog (**18a**) giving long chain esters (**21a** and **21b**) [24].

### 2.9. Synthesis of Monocyclic Phosphate Analogs

Spilling et al. expanded the structural variation in monocyclic analogs of cyclophostin and the cyclipostins with the synthesis of monocyclic phosphates and α,α-difluoro phosphonates [25]. To prepare the phosphate, *t*-butyl acetoacetate was alkylated with iodide (**22**) to give β-ketoester (**23**) (Scheme 6). An important feature of this synthesis is the selection of the *tert*-butyl ester, which minimizes the risk of lactonization upon deprotection of the PMB to reveal the alcohol. Reaction of the β-ketoester (**23**) with dimethyl chlorophosphite, followed by oxidation of crude material with I_2_ and methanol gave enolphosphate (**24**). The PMB ether protecting group was removed with DDQ to give alcohol (**25**). Demethylation and cyclization using 1-mesitylene-sulfonyl-3-nitrotriazole (MSNT) gave monocyclic *tert*-butyl ester (**26**). Cleavage of the *tert*-butyl moiety with TFA in anhydrous conditions was surprisingly effective and is a testament to the stability of the enolphosphate bond. The resulting carboxylic acid was treated with TMSCHN_2_ to give cyclic phosphate methyl ester cyclophostin analog (**27**). Trans-esterification gave the hexadecyl ester cyclipostin P analog (**28**).

### 2.10. Synthesis of α,α-difluoro Phosphonate Analogs

The (allyl-difluoro) phosphonate (**30**) was prepared by reaction of the cuprate of diethyl (bromodifluoromethyl) phosphonate (**29**) with allyl bromide (Scheme 7). Cross metathesis with methyl acrylate using Hoveyda–Grubbs II catalyst gave unsaturated ester (**31**), which was hydrogenated to the saturated ester (**32**). Formation of an enolate, trapping with acetic anhydride and hydrolysis of the crude product gave the β-ketoester (**33**). Selective de-ethylation was accomplished by treatment with NaI in refluxing acetonitrile. Cyclization with MSNT, produced cyclic α,α–difluorophosphonate cyclophostin analog (**34**). Trans-esterification using hexadecyl iodide gave the cyclipostin P analog (**35**). The difluoro enolphosphonates (**34** and **35**) were considerably less stable than the corresponding phosphonates and phosphates, resulting in lower yields after isolation by chromatography.

### 2.11. Biological Activities

Synthetic (±) cyclophostin (**2a**) inhibited human AChE with IC_50_ of 45 nM [11,25]. This compares well to the inhibition of AChE from two different insects by natural cyclophostin (IC_50_ of 0.76 and 1.3 nM) [5], especially allowing for species difference. The diastereoisomer of cyclophostin (**2b**) was similarly active with IC_50_ of 40 nM. The bicyclic phosphonate analogs (**19a** and **19b**) were 10^3^–10^4^ less active with IC_50_ of 30 and 3 μM, respectively [20,21]. The monocyclic phosphate (**27**, IC_50_ = 1 μM) and monocyclic phosphonate (**18a**, IC_50_ = 26 μM) were also modest inhibitors [25]. The difluorophosphonate (**35**) was inactive and was also unstable in aqueous buffer, especially at pH 8.0 [25].

Synthetic (±) cyclipostin P (**3d**) inhibited rat HSL with IC_50_ of 25 nM [24], which again compares well with the reported data for natural cyclipostin P (IC_50_ 30 nM) [7]. The diastereoisomer of cyclipostin P (**13**) was 10-fold less active with IC_50_ of 0.42 μM. The bicyclic phosphonate analogs (**20a** and **20b**) were 10^1^–10^2^ less active with IC_50_ of 6.9 and 0.36 μM, respectively [24]. The monocyclic phosphate (**28**, IC_50_ = 60 nM) and monocyclic phosphonate (**21a**, IC_50_ = 0.54 μM) were also good inhibitors of HSL. Again, the difluorophosphonate (**35**) was inactive and unstable [25]. The C5 alkyl substituted monocyclic phosphonates (**18d**–**18h**) proved to be weak inhibitors of HSL with IC_50_ in range of 10 to >60 μM. Cyclophostin was inactive against HSL. It is interesting that the change from C16 alkyl to methyl phosphate switches selectivity from potent HSL inhibitor to potent AChE inhibitor, with little affinity for the other enzyme.

Although, the C5 alkyl substituted monocyclic phosphonates (**18d**–**18h**) were originally designed as inhibitors of HSL, they were shown to have good activity against microbial lipases [22,23]. Furthermore, compounds (**3d**), (**18e** and **18f**) and (**28**) were shown to possess desirable activity against *Mycobacterium tuberculous* (*M.tb.*, MIC_50_ = 0.5–11.7 μM) and other *mycobacteria* [26,27]. Importantly, these inhibitors exhibited very low toxicity towards host macrophages (CC_50_ > 100 µM). Interestingly, data shows that the compounds exhibit two types of antibacterial activity. Some compounds have higher activity against intracellular than against extracellular bacteria. This suggests that there may be different modes of action with each set of compounds. Using activity-based protein profiling, 23 potential target enzymes of compound (**28**), which exhibited the best extracellular anti-tubercular activity, were identified [26,27,28,29,30]. Remarkably, all of the identified proteins were serine or cysteine enzymes; and most of them are involved in *M. tb* lipid metabolism or cell wall biosynthesis. Among them, the antigen 85 complex and TesA, playing key role in mycolic acid metabolism, have been further characterized as targets, and their respective crystal structures in complex with compounds (**18f**) and (**28**), respectively, have been solved [28,29].

Mass spectroscopic and X-ray crystallographic studies with several different enzymes [21,22,26,27,28,29,30] have shown that the enolphosphates (and phosphonates) function by phosphorylation of the active site serine (Figure 5). In some cases, the inactivated enzyme undergoes further chemistry by loss of the β-ketoester moiety [28].

Given all of the data surrounding the cyclipostins and analogs, it is quite likely that the salinipostins act by inhibiting one or more serine hydrolase enzyme critical to plasmodium. The authors [10] suggest that the structure activity profile of the salinipostins resembles that of inhibitors of fatty acid synthase or enzymes responsible for the post translational modification of proteins (palmitoylation). Furthermore, attempts to produce *P. Falciparum* resistant to salinipostin A failed, suggesting these compounds target multiple enzymes as seen with *M.tb*.

## 3. Conclusions

Cyclophostin, the cyclipostins and the salinipostins represent an interesting class of biologically active phosphate containing natural products that have inspired the synthesis of several analogs.

There would appear to be enormous potential for cyclic enolphosphate and phosphonate analogs of cyclophostin (and the cyclipostins and salinipostins) to become a new general class of serine hydrolase inhibitor. Modifications can be easily introduced in X, Y, Z, R^1^, R^2^, R^3^ and R^4^ (Figure 6), to tailor specificity for a particular enzyme. Indeed, the strategies shown above already allow for the synthesis of a large variety of fully customizable monocyclic enolphosphonates. Furthermore, the methyl phosphonate and phosphates esters can be easily trans-esterified to give long chain ester compounds (ZR^1^ chains, Z = O) analogous to the cyclipostins.

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
