# Peer review of "The Chemistry and Biology of Cyclophostin, the Cyclipostins and Related Compounds"

_molecules, 2019, doi:10.3390/molecules24142579_

Round 1
Reviewer 1 Report
This ms described the synthesis and Biology of Cyclophostin, Cyclipostins and Related Compounds. Comments are:
1) Substituents in Figure 2 need to show a dash line so the readers know where the attachment point is.
2) Page 3 line 54 Further studies also showed that cyclipostins possess anti-mycobacterial activity.
Comment: What’s the mechanism of action?
3) Page 3 line 59 Salinipostin A is a potent growth inhibitor of Plasmodium Falciparum (EC50 = 50 nM), the causative agent of malaria.
Comment: What’s the mechanism of action?
4) The synthesis shown in Scheme 3 (The synthesis of the salinipostins) is the same as the ones shown in Schemes 1 and 2, so what’s the point of repeating?
5) Other that targeting AChE, what other biological functions do these molecules have? For example, do they inhibit proteases?
6) The author should discuss the pros and cons of different classes of molecules in terms of their biological activity and selectivity.
7) The biological activity of these molecules needs to be discussed in the context of high resolutions structures if they are available.
Author Response
1) Substituents in Figure 2 need to show a dash line so the readers know where the attachment point is.
The point of attachment has been indicated by a dot.
2) Page 3 line 54 Further studies also showed that cyclipostins possess anti-mycobacterial activity.
Comment: What’s the mechanism of action?
The mechanism of action was not directly discussed in the patent, although reference was made to HSL type lipases produced by M.tb. A sentence has been added stating this.
3) Page 3 line 59 Salinipostin A is a potent growth inhibitor of Plasmodium Falciparum (EC50 = 50 nM), the causative agent of malaria.
Comment: What’s the mechanism of action?
The mechanism of action is not known, but the authors suggest that the structure activity profile of the salinipostins resembles that of inhibitors of fatty acid synthase or enzymes responsible for the post translational modification of proteins (palmitoylation). Furthermore, attempts to produce P. Falciparum resistant to salinipostin A failed, suggesting these compounds target multiple enzymes as seen with M.tb. Two sentences stating this have been added to section 2.11 (biological activity).
4) The synthesis shown in Scheme 3 (The synthesis of the salinipostins) is the same as the ones shown in Schemes 1 and 2, so what’s the point of repeating?
The two schemes have been merged
5) Other that targeting AChE, what other biological functions do these molecules have? For example, do they inhibit proteases?
Some of the known enzyme targets (HSL, TesA, Lip G, Antigen 85C) were discussed in section 2.11. However, identification of the target enzymes is ongoing and so the question cannot be answered with any certainty.
6) The author should discuss the pros and cons of different classes of molecules in terms of their biological activity and selectivity.
There is not enough data to do this with any confidence.
7) The biological activity of these molecules needs to be discussed in the context of high resolutions structures if they are available.
This was already included in the original manuscript (section 2.11).
Reviewer 2 Report
The paper must be rewritten because is not well organized.
A review has an abstract, an introduction, a critical presentation of the literature, some conclusions and references. So please write in it accordance.
Extend introduction Chapter;
Remove headings: 2. Results; 3. Discussion;
Move the text from Discussion chapter to introduction chapter;
Add: Conclusion(s).
Author Response
A review has an abstract, an introduction, a critical presentation of the literature, some conclusions and references. So please write in it accordance.
Extend introduction Chapter;
An additional sentence has been added.
Remove headings: 2. Results; 3. Discussion;
The results heading has been removed and the whole of the literature review has been placed under discussion.
Move the text from Discussion chapter to introduction chapter;
This doesn’t make sense. This discussion flows from the information in the literature. It would be out place in the introduction.
Add: Conclusion(s).
A conclusion has been added.
Reviewer 3 Report
The review on cyclophostin, cyclipostins and their related derivatives was written concisely with comprehensive coverage of the latest literature. Can be published in its present form.
Author Response
No changes requested. Thank you.
Round 2
Reviewer 1 Report
Comments from the previous round of review were properly addressed.
Reviewer 2 Report
The paper was improved significantly. Accept.